# Solar Tracking Control Algorithm Based on Artificial Intelligence Applied to Large-Scale Bifacial Photovoltaic Power Plants

**DOI:** 10.3390/s24123890

**Published:** 2024-06-15

**Authors:** José Vinícius Santos de Araújo, Micael Praxedes de Lucena, Ademar Virgolino da Silva Netto, Flávio da Silva Vitorino Gomes, Kleber Carneiro de Oliveira, José Mauricio Ramos de Souza Neto, Sidneia Lira Cavalcante, Luis Roberto Valer Morales, Juan Moises Mauricio Villanueva, Euler Cássio Tavares de Macedo

**Affiliations:** 1Renewable and Alternatives Energies Center (CEAR), Electrical Engineering Department (DEE), Campus I, Federal University of Paraiba (UFPB), João Pessoa 58051-900, Brazil; jose.araujo@cear.ufpb.br (J.V.S.d.A.); ademar@cear.ufpb.br (A.V.d.S.N.); mauricio@cear.ufpb.br (J.M.R.d.S.N.); jmauricio@cear.ufpb.br (J.M.M.V.); 2Renewable and Alternatives Energies Center (CEAR), Department of Renewable Energy Engineering (DEER), Campus I, Federal University of Paraiba (UFPB), João Pessoa 58051-900, Brazil; micael.lucena@cear.ufpb.br (M.P.d.L.); flavio@cear.ufpb.br (F.d.S.V.G.); kleber.oliveira@cear.ufpb.br (K.C.d.O.); sidneia.cavalcante@cear.ufpb.br (S.L.C.); 3Huawei Digital Power Brazil, São Paulo 04711-904, Brazil; luisr.morales@huawei.com

**Keywords:** solar tracker, Artificial intelligence, backtracking, diffuse irradiance, machine learning, bifacial solar panels

## Abstract

The transition to a low-carbon economy is one of the main challenges of our time. In this context, solar energy, along with many other technologies, has been developed to optimize performance. For example, solar trackers follow the sun’s path to increase the generation capacity of photovoltaic plants. However, several factors need consideration to further optimize this process. Important variables include the distance between panels, surface reflectivity, bifacial panels, and climate variations throughout the day. Thus, this paper proposes an artificial intelligence-based algorithm for solar trackers that takes all these factors into account—mainly weather variations and the distance between solar panels. The methodology can be replicated anywhere in the world, and its effectiveness has been validated in a real solar plant with bifacial panels located in northeastern Brazil. The algorithm achieved gains of up to 7.83% on a cloudy day and obtained an average energy gain of approximately 1.2% when compared to a commercial solar tracker algorithm.

## 1. Introduction

Photovoltaic systems are becoming increasingly important due to their potential to provide clean and renewable energy [1]. In addition to their capacity to mitigate climate change by reducing greenhouse gas emissions, photovoltaic systems offer numerous other benefits. One promising development in the field of photovoltaic systems is the use of bifacial photovoltaic panels, which can capture energy from both sides of the panel, thereby increasing their efficiency [2]. Bifacial photovoltaic panels have the potential to increase energy yield by up to 25% compared to traditional panels, according to the National Renewable Energy Laboratory (NREL). Additionally, bifacial panels are more durable and resistant to weather and mechanical stress, making them suitable for harsh environments. These advantages have increased the use of bifacial panels in large-scale installations, particularly in regions with high solar irradiance, such as the Middle East and North Africa [3]. However, solar trackers are another technology that has gained increasing attention for further improving the efficiency of photovoltaic systems.

Solar trackers are devices that orient photovoltaic panels toward the sun to maximize energy capture. By tracking the sun’s movement across the sky, solar trackers can increase the amount of energy captured by photovoltaic panels by up to 25–40% compared to fixed-tilt systems [4]. This technology has become more affordable in recent years, making it increasingly popular for large-scale installations. Solar trackers are especially vital in areas with high variability in solar irradiance throughout the day or the year, such as regions near the equator, like Brazil. Additionally, the use of solar trackers can also help reduce the land footprint of photovoltaic systems by allowing more energy to be captured on a given area of land. Overall, combining bifacial panels and solar trackers can significantly increase the efficiency and energy yield of photovoltaic systems, making them an increasingly important technology for meeting our energy needs in a sustainable and environmentally friendly way.

This work aims to present a new artificial intelligence-based algorithm applied to solar trackers that consider bifacial panels to enhance energy generation. The algorithm primarily focuses on exploiting diffuse irradiance and improving the backtracking algorithm. The algorithm presented in this work also improves the performance of solar trackers by considering the unique characteristics of bifacial panels and the physics characteristics of the solar plant, such as the GCR and Albedo. Moreover, the algorithm was validated in a large-scale solar power plant, where solar trackers can adjust the orientation of bifacial panels to optimize energy capture throughout the day.

Thus, the main contributions are as follows:An algorithm able to increase the energy gain during cloudy days;An algorithm able to increase the energy gain during the backtracking period;The development of a simple methodology to build the IA-based algorithm for solar plants to be used worldwide;An algorithm able to consider different scenarios and particularities of a solar plant.

## 2. Related Works

Solar trackers (STs) can currently be categorized into two primary groups, which are distinguished by their movements: single-axis trackers (rotating around a single axis) and double-axis trackers (rotating around two axes).

In addition, advancements in the manufacturing of PV panels and concentrating solar power (CSP) systems, as well as the use of advanced computer technology and reliable control systems, have created new research opportunities focused on optimizing the ST design and operational algorithms.

Thanks to these technical advancements, these devices achieve remarkable increases in efficiency [5,6,7] on both clear and cloudy days [8,9].

In [6], the authors compared fixed flat and single-axis ST PV systems. Moreover, single-axis STs can quickly increase their energy production by 12–20% compared to fixed flat PV systems. An energy increase of up to 30% can still be achieved if the tracker is further optimized [10,11].

In [12], STs were categorized based on four characteristics: active or passive; single axis or two axes; open or closed-loop control systems; and chronological or sensor-based tracking strategies.

Passive tracking systems do not employ any mechanical device controlled by an electrical/electronic system to track the best position concerning the sun.

On the other hand, active tracking systems use a combination of sensors and motors that can adjust the panels to face the direction of maximum light intensity. This technique helps to improve the system’s efficiency, as the amount of electrical energy generated is directly proportional to the incident sunlight energy. Active tracking systems usually consist of a pair of photosensitive elements that can adjust the level of an electrical parameter based on the intensity of the incoming sunlight radiation.

An alternative active tracking approach involves positioning the system based on the theoretical position of the sun. By performing calculations based on the date and time, it is possible to determine the exact location of the sun at that particular moment. The tracking system can then perform these calculations and activate the actuators to align the panels with the sun’s position [13].

According to Helwa et al. [14], the vertical axis system is considered the most effective among these types of systems. These systems are efficient, owing to their simple construction and easily manageable control system.

Automatic solar tracking systems (ASTSs) can position solar power systems to optimize energy absorption by orienting them perpendicular to incoming solar rays. These systems usually consist of components, including transmission mechanical drive subsystems, electric motors, sun position sensors, solar position algorithms, control units, and limit switches.

In the literature, it is also possible to find some works that have attempted to study methods to increase the efficiency of photovoltaic systems using solar trackers. In [15], a study was presented using pyranometers to measure solar irradiance on cloudy days. The study found that positioning the panels horizontally on such days is more advantageous than having them follow the sun. By comparing two pyranometers—one installed horizontally and one tracking the sun—it was observed that the pyranometer in the horizontal position collected 50% more irradiance than the one tracking the sun during a cloudy day.

In [16], the potential for irradiance collection of solar panels by moving them to the horizontal position during cloudy and rainy days was studied within the European geographical context. Using data from the European Baseline Surface Radiation Network (BSRN) and available irradiance transposition models in the literature, it was observed that in the more northern regions of Europe, there was a potential to increase irradiance collection by up to 3.0% per year, with records showing an increase of almost 20% on strongly cloudy days, compared to the strategy of merely following the sun’s position. Additionally, the authors proposed two algorithms to capitalize on this potential gain on cloudy days, one of which could increase irradiance collection by 2.5% per year, according to the mathematical analyses performed.

In [17], the potential gain in irradiation collection was analyzed through simulation and mathematical calculations using irradiation transposition models, considering energy collection in both monofacial and bifacial photovoltaic panels. In this study, the authors analyzed data from 61 weather stations, comparing two tracker control strategies: following the sun and tracking the best orientation, which calculates the optimal angle via “brute force” through knowledge of direct, diffuse, and ground-reflected irradiance. The analyses estimated a gain of more than 3.9% in the annual irradiation collection for bifacial photovoltaic modules equipped with single-axis solar trackers.

In [18], a comparative analysis was presented through simulation between three backtracking algorithms for solar power plants installed on sloping surfaces. The backtracking algorithms decrease shading between rows of photovoltaic panels that can occur at sunrise and sunset. With their proposed algorithm, the average annual energy gains in a relief context were between 7 and 8%.

In [19], a study similar to the one developed in [17] was performed, but considering the geographical context of the United States, evaluating the potential for improvement in irradiance collection based on direct and diffuse irradiance. Similarly, in [17], a “brute force” strategy was used to calculate the optimal positioning angle of the photovoltaic panels given the position of the sun and the irradiance at the time. A potential gain of up to 1% was found, considering hourly weather data extracted from the typical meteorological year (TMY) for the United States region.

A study with practical results was presented in [20], where the authors developed a prototype solar tracker with three photovoltaic panels: one higher-capacity photovoltaic panel coupled with a solar tracker oriented according to the energy performance of two other smaller solar modules—one that follows the sun’s position and another that remains horizontal. Between these two smaller modules, the one generating more energy would dictate the position of the larger module. As a result, at a specific time on a cloudy day, the proposed algorithm was able to generate 18% more energy.

In another practical study, ref. [21] compared a new approach to backtracking algorithms for optimizing power generation during sunrise and sunset. The proposed algorithm improved the yearly average energy generation performance by about 2.42% over the commercial backtracking algorithm, solely by improving the positioning of the panels during these periods of the day.

Therefore, according to all the works analyzed, the two main elements that can be optimized refer to the positioning of the photovoltaic panels during cloudy and rainy days and also focusing on the backtracking period. On the other hand, it was noted that only a few studies presented practical results of their analyses.

For these reasons, this paper proposes presenting and detailing a new algorithm for solar trackers based on artificial intelligence to address all the cited deficiencies in solar tracker systems, such as harnessing diffuse irradiance on cloudy days and improving backtracking algorithms; moreover, the algorithm’s performance will be validated in the field with bifacial solar panels. In this way, an energy performance comparison of the proposed algorithm with traditional solar tracker algorithms that only track the sun and use a basic backtracking algorithm will be presented in the results.

## 3. Materials and Methods

In this section, details about the adopted scenario for the case study of the comparison between the proposed intelligent algorithm for solar trackers and the conventional algorithm largely used in the market are presented. In addition, the equipment involved in the process is detailed.

Moreover, this section will outline the procedure used to develop the algorithm based on artificial intelligence (AI), presenting the methodology adopted from data collection through simulation to the validation process for algorithm development. Additionally, it will discuss the performance improvement process of the proposed solution, considering the various techniques used, the criteria for choosing the variables, and finally, the considerations made to quantify the energy gain of the developed algorithm according to the weather classification.

### 3.1. Materials

In this subsection, a brief overview of solar tracker technologies is presented. In addition, the equipment used in the proof of concept scenario is detailed, listing its functions and the communication between them.

To summarize, the developed AI-based solution operates as a plug-in at the control and field levels of the automation pyramid, positively impacting the generation results, as shown in the following sections. As seen in Figure 1, for this case study, the developed algorithm operates at the same level as the control layer, interacting with the field process (weather station and solar trackers) through the Modbus TCP protocol, sending the calculated angle to the solar trackers based on the variables read from the sensors.

#### 3.1.1. Solar Tracker

The tracker system utilized was the STI-250 from STI Norland, which is predominantly used in commercial solar plants in Brazil. This system is categorized as a one-axis solar tracker. The panels were installed facing north (for countries below the equator line), and the solar tracker system tracks the sun from east to west.

To follow the sun, an astronomical algorithm named SPA is used to calculate the sun’s position. NREL developed this algorithm, which STI Norland has implemented in their solar tracker systems. In addition to tracking the sun, the start of the morning and the end of the afternoon mark the backtracking period, when the tracker rows may shadow each other. STI Norland uses a proprietary algorithm to avoid shadowing the panels during this period.

This article labels the junction of these two algorithms as a “commercial solution”. As such, it is used as a reference to compare the developed solution.

Moreover, the trackers’ architecture is based on a tracker control unit (TCU) per tracker row. The unit can individually control the motor that regulates the panel’s position and guarantees the system’s operation.

Furthermore, a network controller unit (NCU) is used for larger solar plants to operate multiple TCUs. The communication between NCUs and TCUs is made through the ZigBee protocol, and the NCU can be accessed by the Modbus TCP communication protocol.

The tracker’s standard operation is the “automatic mode”, which uses the STI Norland backtracking algorithm with the SPA to calculate the sun’s position. To adopt the developed solution in the field, the trackers are set to “manual mode” through the NCU using the Modbus TCP communication protocol. Then, the optimized angles calculated by AI are written in the STI tracker system registers.

#### 3.1.2. Test Environment Architecture

The tracker’s standard operation is an “automatic mode”, which uses the STI Norland backtracking algorithm with the SPA to calculate the sun’s position. The trackers are set to “manual mode” through the NCU using the Modbus TCP communication protocol to use the developed solution. Then, the optimized angles calculated by AI are written in the STI tracker system registers.

To minimize the influence of external variables, these two inverters are located near the weather station and have similar physical characteristics. The plant parameters are listed in Table 1.

As shown in Table 1, each inverter is connected to six solar tracker rows, with 360 solar panels per set. This distribution is shown in Figure 2.

In addition, it is necessary to receive and send data to the equipment to operate the AI solution. The communication protocols used in the implementation are listed in Table 2.

The data are read from and written directly to the solar tracker NCU by the respective TCUs. System-generated energy and instantaneous power are obtained through the Huawei SmartLogger, which is connected to the inverters; moreover, weather data are collected from the weather station.

Lastly, some solar plant parameters are listed as follows:Pitch: 5 m;Ground coverage ratio (GCR): 0.5;Albedo: 0.25.

### 3.2. Methods: Developing the Intelligent Algorithm

The procedure presented in Figure 3 is followed for the algorithm development. One of the prerequisites for developing the intelligent algorithm is the simulation of the optimal behavior according to the current weather situation. Thus, the first step of the methodology is to download the weather data to use the NREL (National Renewable Energy Laboratory) satellite database; concerning the simulation environment, the pvlib package [22] available in Python and Matlab was chosen; finally, the algorithm training follows with the optimal angles generated in step two.

So, as the first step, the NREL database was chosen after analytical research on the other available climate databases, taking into account some criteria:Free access: Not considered a mandatory prerequisite but is seen as a positive point;Data availability for the study region: The case study was conducted in a relevant environment in the northeast of Brazil. Therefore, the availability of data for the region is a mandatory prerequisite;Availability of climate data, mainly the DNI, DHI, and GHI variables: For the simulation of optimal behavior, it is necessary to have these three variables available as inputs to the adopted irradiance model;The smallest sampling interval: It is desired that the algorithm operates in accordance with weather changes and the positioning of the sun. So, it is essential that the intervals of the training data are as short as possible; data were obtained at intervals spaced every thirty minutes.

The NREL climate data coverage can be seen in Figure 4, which shows that, depending on the region, data have been available since 1998 with ten-, thirty-, and sixty-minute sampling intervals. Among the available data, to mention just a few, are GHI (global horizontal irradiance), DNI (direct normal irradiance), DHI (diffuse horizontal irradiance), wind speed, wind direction, temperature, precipitation, and atmospheric pressure, meeting the minimum requirements for available variables.

The next step involved simulating the optimal behavior of a solar tracker, given the weather conditions and the characteristics of the solar plant. Python was the programming language used, and pvlib was chosen after a comprehensive review of Python packages for modeling photovoltaic systems. Its selection was based on the extensive availability of documentation and a wide variety of mathematical models for simulating a solar plant. These models include controls for trackers, the absorption of irradiance by mono/bifacial panels, and electrical modeling, among others.

In this sense, the main function used was the “infinite sheds model” [23,24], which is an irradiance transposition model that takes into account the main variables that influence a solar plant, such as the following:Height: height of the panel relative to the surface;Pitch: distance between rows of panels;Sun’s zenith and azimuth;Tracker orientation angle;Width solar panel;Bifaciality;GHI, DNI e DHI;Albedo.

Thus, keeping all parameters constant and varying only the orientation angle of the trackers, it was possible to determine which angle absorbs more irradiance for each specific weather situation. In Figure 5, it is possible to observe a comparison between the angles that collect more irradiance on a sunny day and another on a cloudy day. The black line in the plot represents the optimal angle to absorb more irradiance. It can be seen that following the sun’s position is the best alternative on sunny days, as the DNI is higher, making it better to align with the sun’s normal angle. However, this is not the most effective strategy on cloudy days, when the best angles are closer to the horizontal position (0°), due to the GHI being higher than the DNI on such days. This observation corroborates literature studies indicating that this position is more advantageous as it takes advantage of the absorption of diffuse irradiance.

As mentioned earlier, it is possible to simulate the conventional positioning of commercial solar trackers using pvlib. This library utilizes the SPA (solar position algorithm) [25] to calculate the sun’s position and subsequently determine the position that minimizes the difference between the PV panel surface normal and the sun’s position. When generating a frequency histogram of each angle’s occurrence, a uniform distribution can be observed, as shown in Figure 6.

On the other hand, when comparing the frequency of occurrence for each angle by calculating the “infinite sheds model”, there is a higher frequency for the positioning in the horizontal position, as seen in Figure 7. The frequency of positioning of the photovoltaic panels near the horizontal position is higher, indicating that the algorithm’s opportunities for improvement are concentrated during cloudier days.

Finally, in the last step, the focus was on developing an AI-based solar tracker algorithm. This task demanded more time because many machine learning topologies are known in the literature, resulting in numerous possibilities for creating a solar tracker algorithm with the proposed purpose. Therefore, the authors chose only four powerful techniques used in regression problems to analyze their performance as a solar tracker algorithm and save time.

To decide which AI-based technique would be the most suitable to act as a solar tracker algorithm, a comparison between four machine learning/deep learning techniques was made, analyzing the performance of each one about the optimized angles calculated. These are as follows:Neural network of the LSTM (long short-term memory) type;Neural network of the MLP (multi-layer perceptron) type;DT (decision tree);RF (random forest).

Before evaluating the performance of each one, the dependence level between the input and output variables was studied using mutual information estimation [26], i.e., the more significant the dependence, the higher the value of this index. In this study, the dependency between all input variables concerning the optimal angle of reference was evaluated, and these variables were chosen as input variables to be applied to machine learning techniques. The variables that scored higher than 0.3, as presented in Table 3, were selected to filter out those that could have little influence.

The current timestamp was used in trigonometric calculations to create the variables *Hour_sin* and *Hour_cos*. This operation was adopted because azimuth and zenith are also trigonometric variables that could synergize more effectively during AI model training. The mathematical calculations applied are detailed in Equations (Equation 1) and (Equation 2), where *t* is the index representing the current time instant, *h* is the current hour, and *m* is the current minute.
(1)$$Hour\_sin\left(t\right)=sin\left(\frac{2\cdot \pi \cdot (h_{t}\cdot 60+m_{t})}{1440}\right)$$
(2)$$Hour\_cos\left(t\right)=cos\left(\frac{2\cdot \pi \cdot (h_{t}\cdot 60+m_{t})}{1440}\right)$$

With the definition of the variable selection criterion established, the performance evaluation of each regression technique was conducted using MSE (mean squared error), MAE (mean absolute error), and ME (max error) as metrics to determine the best technique. Furthermore, parametrizations were carried out for each regression technique to optimize performance, such as altering the number of neurons and layers in neural networks and testing different numbers of trees in the random forest. After all parametrization tests for each artificial intelligence topology were completed, the random forest emerged as the most accurate technique across all metrics adopted, as shown in Table 4.

Once the random forest was identified as the AI technique that delivered the best results according to the criteria used, further efforts were made to improve the algorithm’s performance. The first step involved evaluating its performance based on the combination of available input variables. Out of the 11 available variables, 1024 combinations were tested. The performance of the top 7 combinations was very similar, with differences only in the third decimal place, as seen in Table 5.

Thus, following the results obtained in the created ranking, we decided to use the best-ranked combination of input variables in the RF model, as shown in the model schematic presented in Figure 8. Up until this point, the solar tracker algorithm was composed of an RF model that infers the optimal angle according to the seven variables indicated.

Finally, the last action for improving the algorithm involved studying the moments when it faced the most significant challenges in accurately inferring the correct output angle from its application with a set of validation data in a power generation simulation. The simulations revealed that the most challenging times for the algorithm’s inference were during sunrise and sunset, or in other words, during backtracking.

In this context, the idea was to divide the algorithm into three parts, training three RF models to operate during each period of the day: one for morning backtracking (set from 05:30–07:30), another for “common time” (from 07:30–15:30), and a last one for afternoon backtracking (set from 15:30–17:30). For the remaining times that could still experience sunlight but were not covered in the pre-determined time windows, the SPA algorithm was applied to avoid interrupting the operation. In this way, the algorithm’s operation scheme can be summarized as follows in Figure 9, where the night position denotes the application of the SPA, with the additional step of positioning the trackers at 10°.

The intelligent algorithm, meticulously developed, was poised for field application to validate its efficiency in a real scenario. However, certain practical aspects, such as determining the optimal time for the algorithm to operate on the trackers, are yet to be defined.

As a final step before applying the algorithm in a commercial solar plant, a thorough evaluation was conducted. This included determining the ideal time to act on the trackers, how the weather data (ghi and wind speed) would be used as input for the AI model, and whether the instantaneous value would be used or treated in a specific manner.

For the first evaluation, a large set of variables could be considered to determine the optimal time of action, such as the time the tracker spends to move from one angle to another, the energy cost to move the trackers, or fast variations in weather, to name a few examples. For simplicity, the ideal time to apply the angle inferred by the AI model would be 4 min, as this is the average time it takes for a conventional algorithm to update the tracker’s position by one degree.

For the input data, we chose to use a moving average of 4 min—the same interval used for the algorithm to act—to mitigate the effects of instantaneous weather changes, since the rapid passage of a cloud can lead to incorrect algorithm inferences. Because it cannot predict the exact timing of cloud movements, directly applying the measured instantaneous values could cause the algorithm to position for diffuse irradiance when conditions are more favorable for tracking the sun, and vice versa.

With these two final definitions, the algorithm was ready for its field implementation; the results can be seen in Section 4.

## 4. Results

This section presents the results obtained during the application of the proposed AI-based algorithm at the solar power plant, analyzing its performance across different weather conditions. Furthermore, it compares the performance of two versions of the developed algorithm: the first version with non-tuned backtracking, and the second with tuned backtracking.

### 4.1. Analysis of the Results Infield

This subsection details the process used to calculate the energy generated by the inverters in the case study, highlighting the gains or losses from the developed intelligent algorithm compared to the commercial one.

#### 4.1.1. Historical Difference in the Generation Profile

Three months of average power data from the inverters connected to each set of trackers were obtained, with a five-minute resolution when the commercial solution was operating in the photovoltaic plant.

The average electric power generation profile for each month and all three months combined was calculated to account for the physical differences between the positions of tracker groups and their impacts on the gains from the algorithm’s performance. The historical hourly power difference profile between inverters can be observed in Figure 10 where the red color means that the inverter A generates less than the B, and the green indicates the opposite.

Therefore, considering the average hourly energy from these three months of data, the difference between inverters A and B is depicted in Figure 10. This curve represents the difference (calculated as a subtraction) in the average hourly energy between inverters A and B. It can be observed that inverter B generates more energy than inverter A (on average) until approximately solar midday (around 11:30). However, after this point, the trend reverses, with inverter A generating more energy than inverter B. This variation is attributed to the different slopes on the trackers connected to each inverter.

Following the historical difference studied, the average daily energy difference between the two inverters was approximately 0.422% for inverter B in relation to inverter A. Hence, inverter A historically generates less energy; it was chosen to evaluate the actual impacts of the developed algorithms. Thus, any increase in energy generation by this inverter can be attributed to the optimized angles applied by the algorithms.

#### 4.1.2. Energy Attenuation Calculation Method

The energy generation difference between the study case scenarios is smooth. However, to obtain a more accurate comparison of the algorithms, it was necessary to calculate an average difference between the inverters and then use this difference in the daily energy calculations.

The energy per hour was collected in the implementation according to Equation (Equation 3). *E* is the stored energy per hour, Δt is the time variation in seconds, and $P_{inst}$ is the measured instant power. The constant value of 3600 corresponds to the energy collection period of 1 h since the unit used is seconds.
(3)$$E=\sum (P_{inst}\cdot \frac{\Delta t}{3600})$$

On the other hand, the historical mean energy per hour is calculated, representing the generation energy difference between inverters A and B. In this sense, the applied correction in the generated energy corresponds to Equation (Equation 4), where the adjusted energy calculation is used, considering the historical difference between the inverters.
(4)$$E_{A_{adjusted}}=E_{A_{measure}}+\left(E_{B_{historical}}-E_{A_{historical}}\right)$$
where $E_{A_{historical}}$ is the historical mean energy per hour for inverter A, $E_{B_{historical}}$ is the historical mean energy per hour for inverter B, $E_{A_{measure}}$ is the measured energy from inverter A, and $E_{A_{adjusted}}$ is the adjusted energy from inverter A.

Therefore, to evaluate the energy gains obtained with the developed algorithm concerning the commercial algorithm, the calculated daily energy was obtained by summing the calculated energy per hour and considering the energy generation ratio between the two inverters, represented in Equation (Equation 5).
(5)$$\Delta E\left(\%\right)=100\%\cdot \frac{E_{Adaily}-E_{Bdaily}}{E_{Bdaily}}$$

ΔE(%) is the percentage energy difference between the inverters, $E_{Adaily}$ is the daily stored energy in inverter A, and $E_{Bdaily}$ is the daily stored energy in inverter B. So, values of ΔE(%) greater than zero indicate a generation gain for the developed algorithm, whereas negative values represent a loss for the developed algorithm compared with the commercial algorithm.

#### 4.1.3. Weather Classification Method

The method used to classify weather conditions is based on analyzing sky conditions. This analysis was implemented by calculating the Kt, which is the ratio of global to atmospheric radiation as described and used in [27]. The cloud effects on solar radiation were measured based on sky conditions and categorized into four Kt intervals, as specified and applied in [28].

The specified intervals to measure the daily weather classification are described in Table 6. Kt ≤0.55 values represent weather with a predominance of diffuse horizontal irradiance, while Kt ≥0.65 indicates a predominance of direct normal irradiance.

The calculation of Kt is a common alternative used for climatic classification studies, adopted in recent works, such as in [29], where the authors consider sky conditions for weather characterization.

### 4.2. AI-Based Algorithm with Non-Tuned Backtracking

To validate the performance of the AI algorithm, the overall generation gain of the algorithm will be shown, followed shortly by the results according to the weather. This algorithm was named “non-tuned backtracking” because after the results were collected in the first period of analysis, adjustments were made to the backtracking phase of the operation to increase the energy gains.

#### 4.2.1. Solution Overall Performance

The behavior of power generation considering the period from 19 October 2022 to 18 November 2022 is the operating period of version 1. It is shown in Figure 11, where the green/red columns mean the days when the developed algorithm produced more/less energy than the commercial algorithm. The weather profile for each day is indicated in the upper part of the figure, based on the methodology described in Section 4.1.3, where more clouds signify a tendency toward cloudier weather.

The proposed solution achieved favorable generation rates on almost all days, except for five days when the developed solution generated less energy than the commercial solution. Despite this, as expected, the algorithm performed better on the other days, especially on partially cloudy and cloudy days, since the developed algorithm is better suited for diffuse irradiance. On 4 November 2022, a cloudy day, an energy gain of 7.83% was achieved, representing the most significant gain of the developed algorithm compared to the commercial algorithm during the entire window application.

#### 4.2.2. Performance by Weather State

A sunny day was selected to begin the updated algorithm’s performance analysis. Figure 12a shows the irradiance profile of that specific day, and Figure 12b presents a comparison between the angles calculated by the two algorithms under study. The developed algorithm exhibited a stepped profile, which resulted from being trained on a dataset sampled every 30 min. In contrast, the algorithm must act every 4 min, leading to some loss of precision. However, the trackers controlled by the developed algorithm followed the Sun’s movement similarly to the commercial algorithm, making it the best strategy for days with high irradiance. On this day, the energy gain was 0.40%, due to the positive aggressiveness in the afternoon backtracking.

For a partially sunny day, shown in Figure 13, it is notable that the irradiance variations throughout the day are accompanied by variations in angulation that are more pronounced than previously observed on a sunny day. Even so, the algorithm presents a behavior close to the commercial algorithm. It is worth noting that, in this type of climate, it is already possible to observe a tendency to seek the horizontal position in moments of sudden drops in the measured irradiance. For this day, the energy gain was 0.51%.

On a partially cloudy day, as presented in Figure 14, the algorithm began to demonstrate behavior with more angular variations, showing a strong preference for the horizontal position during periods of low global irradiance. It is also worth noting that the sharp variations in the angulation correspond with the irradiance peaks. As a result, on this day, the energy gain was 2.98%.

On a cloudy day, the developed solution behaved in a similar way to that previously analyzed for a partially cloudy day; that is, it is possible to observe substantial variations between the angles adopted by the developed algorithm and the commercial one, with emphasis on an increasing tendency toward a horizontal position. Still, in this aspect, the irradiance peaks also coincide with the variations observed in Figure 15. As a result, for this day, the energy gain was 7.83%, representing the best energy gain of the analyzed period.

The developed algorithm was evaluated over 29 days and showed that the energy gain was more significant on cloudy days, with an energy gain superior to 5%. However, on sunny days, it performed comparably to the commercial algorithm. Considering the first version of the developed algorithm, the total energy gain was 273.90 kWh, with an average performance of 0.73%. In Figure 16, it is possible to see a summary of the average energy gains achieved by the algorithm by weather type during this implementation window.

### 4.3. AI-Based Algorithm with Tuned Backtracking: Update

The pvlib conventional angle calculation was used to improve the intelligent model during the morning backtracking period, as the solution lost energy due to aggressive inferences during this period, as seen in Figure 17.

It was observed that during certain times in the morning backtracking, the AI-calculated angle was more aggressive than expected, resulting in unexpected shadows. Therefore, to enhance the AI algorithm during morning backtracking, a hybrid approach was employed. This approach used both the pvlib calculations and the AI-based calculations, where the maximum angle calculated by pvlib was used to limit the AI-calculated angle. This helped avoid shading by subtracting 3 degrees from the AI’s aggressive angle to check if shading losses could be attenuated. In Figure 18, it is possible to observe the change caused by this action.

#### 4.3.1. Solution Overall Performance

As previously mentioned, the modifications made to the algorithm had a positive effect, reducing energy loss during this period and consequently increasing the overall efficiency of the solution. The behavior of power generation during this version of the algorithm (19 November 2022 to 22 December 2022), the operating period for version 2, is depicted in Figure 19. The proposed solution achieved favorable generation rates on almost all days except two, where the developed solution generated less energy compared to the commercial solution. Despite this, the algorithm performed better on the other days, especially on partially cloudy and cloudy days. On 23 November 2022, a cloudy day, an energy gain of 6.39% was achieved, representing the most significant gain of the developed algorithm compared to the commercial algorithm during this application window.

#### 4.3.2. The Effect of Weather on AI-Algorithm Performance Evaluation

Starting the performance analysis of the updated algorithm on a sunny day, the controlled trackers followed the sun’s movement in a similar way to the commercial algorithm, being the best strategy for days with high irradiance emission, as shown in Figure 20. In addition, the algorithm responded to the few irradiance variations throughout the day. For this day, the energy gain was 0.97%.

For a partially sunny day, shown in Figure 21, it is possible to verify that the irradiance variations throughout the day are accompanied by more pronounced angulation variations than those previously observed on a sunny day. Even so, the algorithm exhibits behavior similar to the commercial algorithm. It is important to note that, in this type of climate, there is a noticeable tendency to seek the horizontal position during moments of sudden drops in irradiance. For this day, the energy gain was 0.64%.

On a partially cloudy day, Figure 22, as in the first version of the algorithm, the intelligent solution presented a strong preference for the horizontal position compared to the angles adopted by the commercial algorithm. It is worth noting that the sharp variations in the angulation are compatible with the irradiance peaks. As a result, for this day, the energy gain was 3.74%.

On a cloudy day, the developed solution achieved its best performance, as with the first version of the algorithm. Its behavior was similar to that on the partially cloudy day, with a strong preference for the horizontal position. Still, the irradiance peaks coincided with the variations observed in Figure 23. As a result, for this day, the energy gain was 6.39%, which is lower than that of the first algorithm version. This difference was due to the fact that the first window analysis encountered cloudier conditions than the second window.

The second version of the algorithm was evaluated over 34 days and demonstrated an energy gain that was more significant on cloudy days, with an energy gain superior to 5%. The total energy gain during the application of this second version was 493.710 kWh, with an average performance of 1.20%. In Figure 24, it is possible to see a summary of the average energy gains obtained by the algorithm concerning weather conditions during this implementation window. Compared to the first version of the algorithm, as shown in Figure 24, it is evident that this update had the desired effect by increasing the energy gains, especially on days with partially sunny and sunny weather.

### 4.4. Solution Overall Performance Evaluation

Table 7 summarizes the energy gain at each interval. The results show that the second version of the algorithm increased energy generation. It improved its performance during the backtracking period, reducing the losses in the first algorithm’s version and maintaining the performance during cloudy days.

## 5. Conclusions

The algorithm was validated in a commercial solar power plant, where the performance of the solar trackers was compared to a reference setup considering a commercial algorithm. The results showed that the algorithm improved the energy generation using bifacial panels by up to 1.2% compared to the reference case. The algorithm also optimized the conventional angles suggested by the traditional solar tracker tracking algorithm, leading to a more precise orientation of the bifacial panels. These results demonstrate the effectiveness of the new algorithm for solar trackers that use bifacial panels.

It is worth noting that the proposed algorithm achieved a smaller performance improvement on sunny days compared to cloudy days relative to the traditional algorithm. This is because both algorithms behaved similarly in the angles sent to the trackers, as the optimal position on these days is to follow the Sun. On the other hand, on cloudy days, the algorithm achieved a performance improvement of 7.83% over the traditional algorithm, demonstrating its potential, particularly on these days.

In conclusion, the development of new algorithms for solar trackers that take into account the unique characteristics of bifacial panels can significantly improve their performance and energy yield. The algorithm presented in this work offers a promising solution for maximizing the energy capture of bifacial panels in solar power plants and has the potential to make a significant and transformative contribution to the development of efficient and sustainable energy systems.

From the algorithm developed in this research, future work is proposed to compare the generation gain performance with that of other power plants and to evaluate scalability with commercial solutions integrated into the cloud, making application easy in different setups of large solar plants.

Furthermore, future work can also study the adaptation of this algorithm to solar plants with non-flat terrain (considering that the backtracking relationship in this type of situation has a different optimal angle profile) and test new algorithms and databases for the development of this solution.

## Figures and Tables

**Figure 1 sensors-24-03890-f001:**
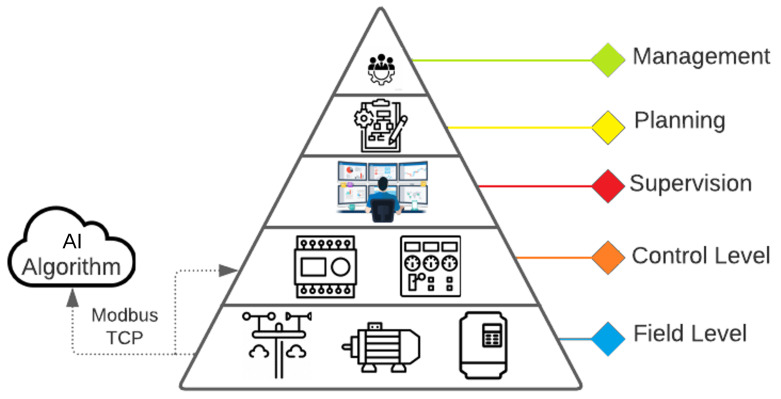
Integration of the AI solution in the automation pyramid.

**Figure 2 sensors-24-03890-f002:**
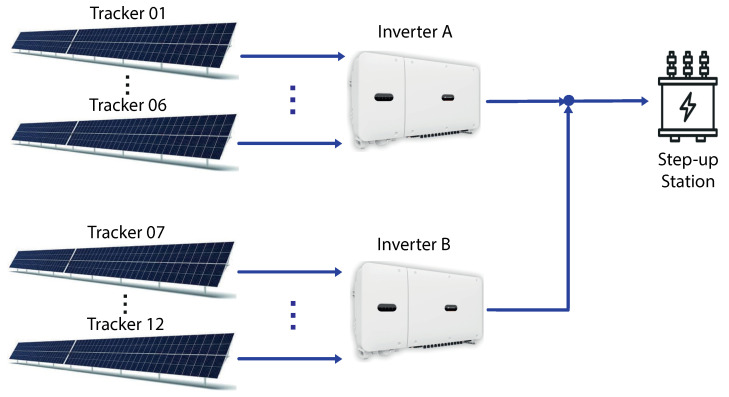
Solar trackers and inverters connection diagram.

**Figure 3 sensors-24-03890-f003:**
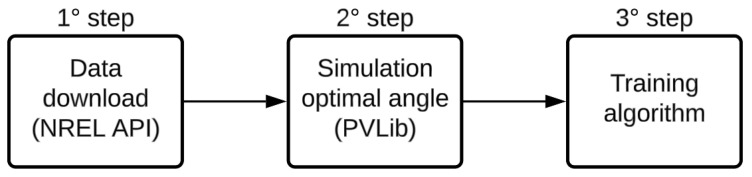
Methodology diagram.

**Figure 4 sensors-24-03890-f004:**
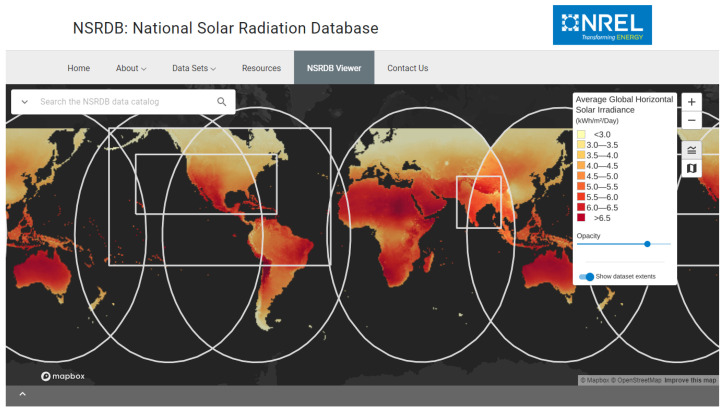
NREL database coverage.

**Figure 5 sensors-24-03890-f005:**
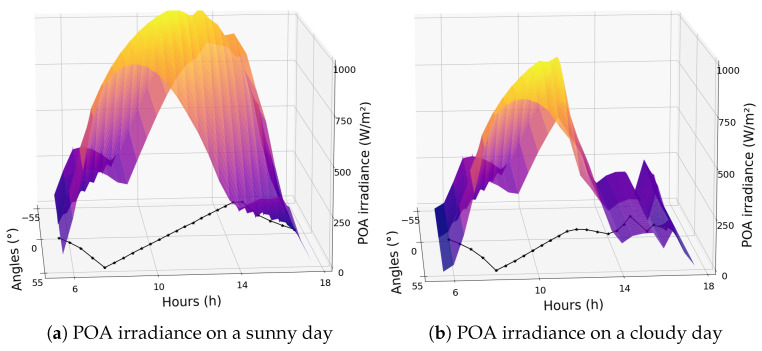
Comparison between the optimal angle on a sunny day (**a**) and a cloudy day (**b**).

**Figure 6 sensors-24-03890-f006:**
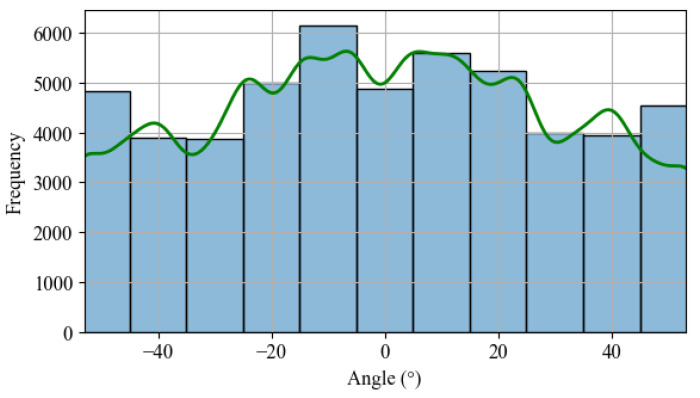
Histogram: angle frequency occurrence based SPA calculation.

**Figure 7 sensors-24-03890-f007:**
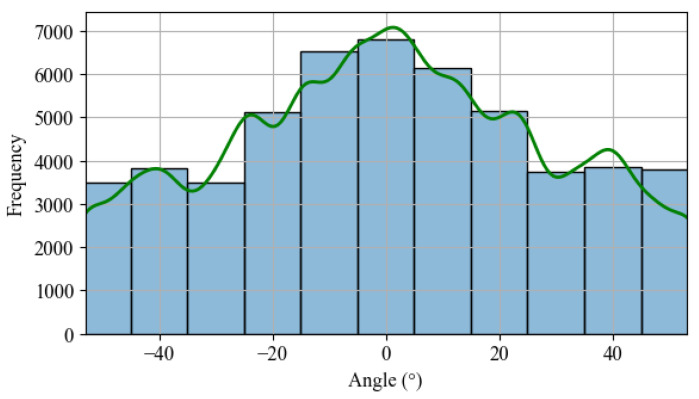
Histogram: angle frequency occurrence based on the “infinite sheds model”.

**Figure 8 sensors-24-03890-f008:**
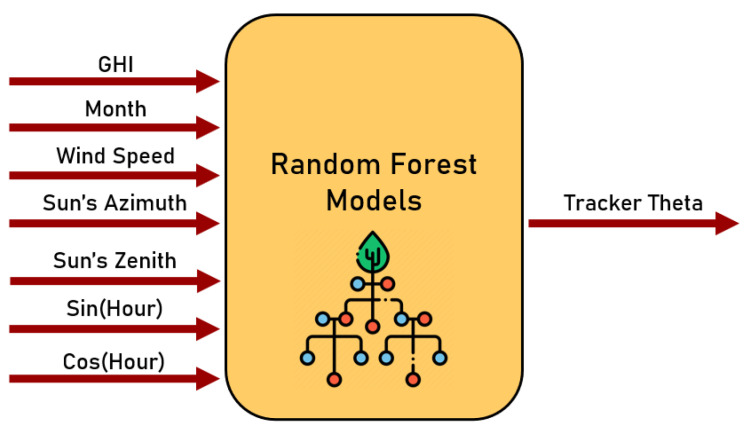
Random forest model’s inputs.

**Figure 9 sensors-24-03890-f009:**
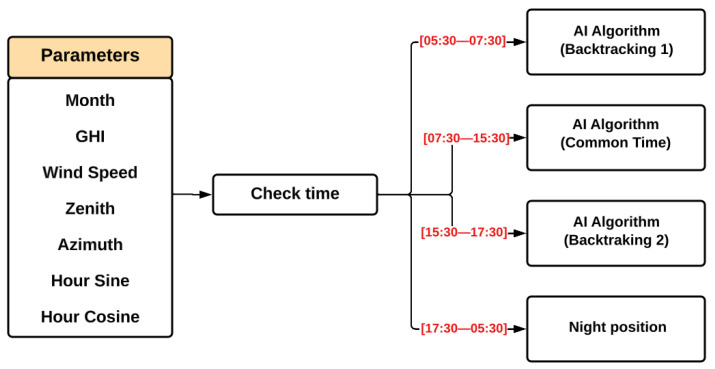
The AI algorithm’s detailed operation process.

**Figure 10 sensors-24-03890-f010:**
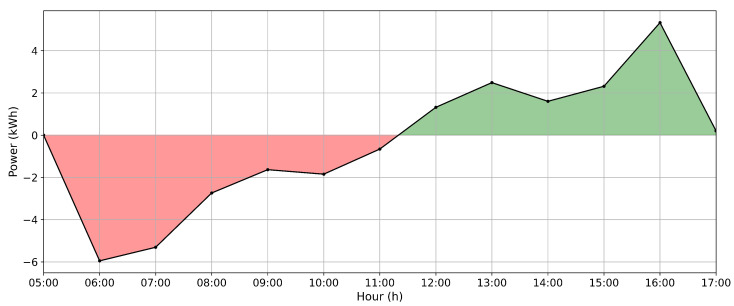
Historical profile: power difference between inverters A and B.

**Figure 11 sensors-24-03890-f011:**
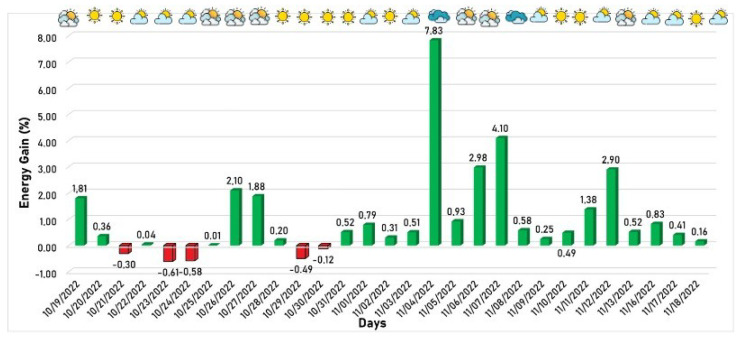
First version of developed algorithm energy gain in relation to the commercial algorithm.

**Figure 12 sensors-24-03890-f012:**
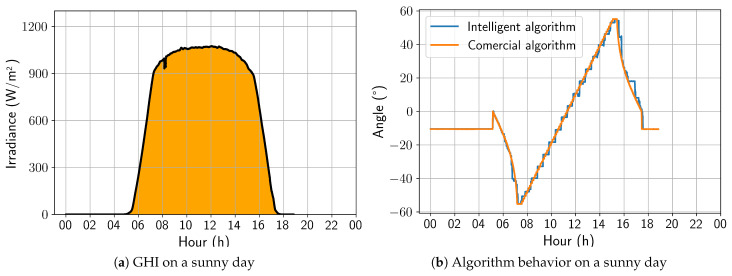
Comparison between intelligent and commercial algorithms for a sunny day; 17 November 2022.

**Figure 13 sensors-24-03890-f013:**
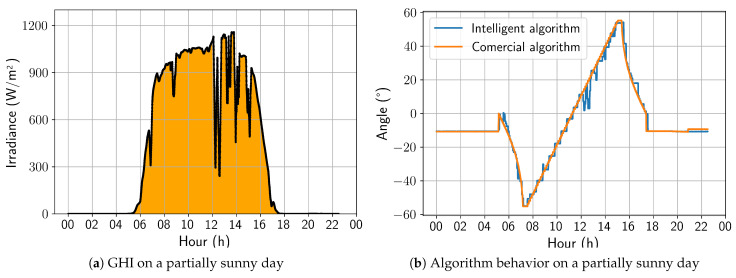
Comparison between intelligent and commercial algorithms for a partially sunny day; 3 November 2022.

**Figure 14 sensors-24-03890-f014:**
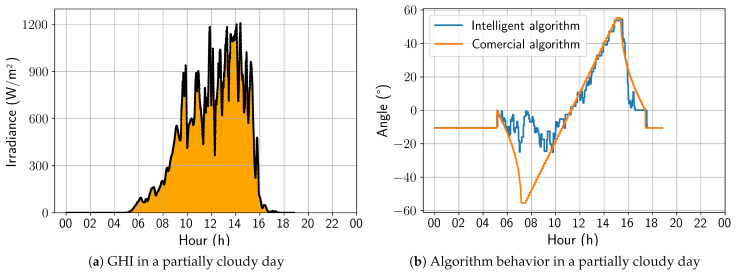
Comparison between intelligent and commercial algorithms for a partially cloudy day; 6 November 2022.

**Figure 15 sensors-24-03890-f015:**
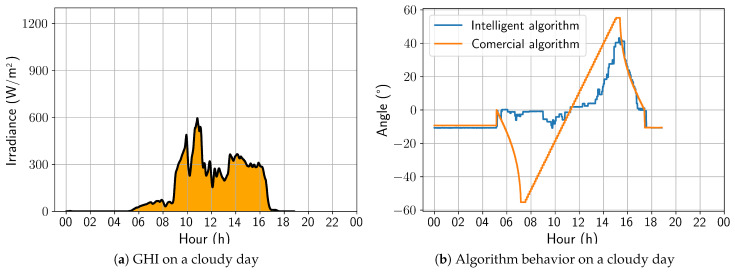
Comparison between intelligent and commercial algorithms for a cloudy day; 4 November 2022.

**Figure 16 sensors-24-03890-f016:**
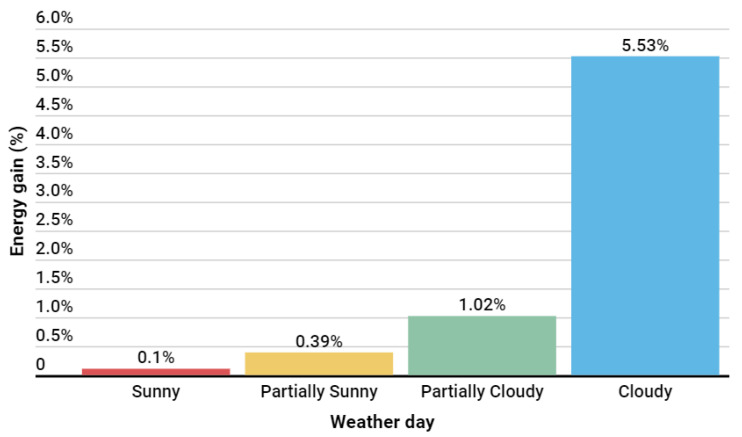
Results by weather situation with the first algorithm version.

**Figure 17 sensors-24-03890-f017:**
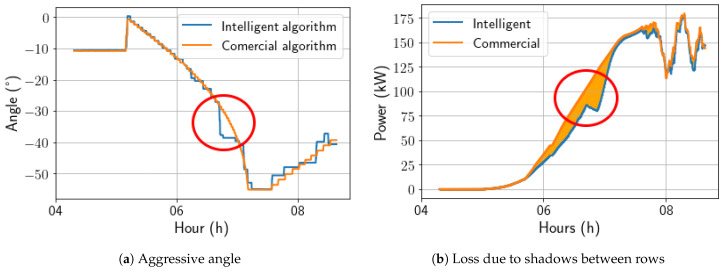
Comparison between intelligent and commercial algorithms before the update.

**Figure 18 sensors-24-03890-f018:**
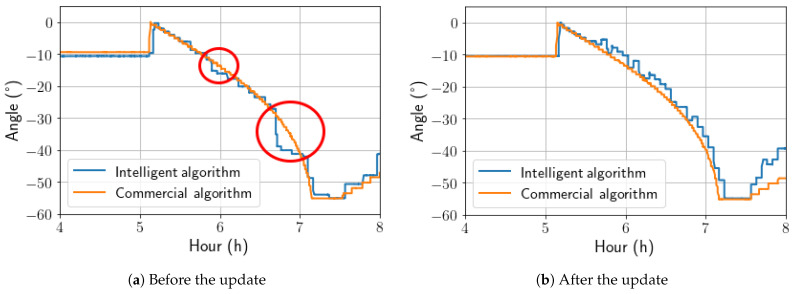
Update in algorithm operation during morning backtracking.

**Figure 19 sensors-24-03890-f019:**
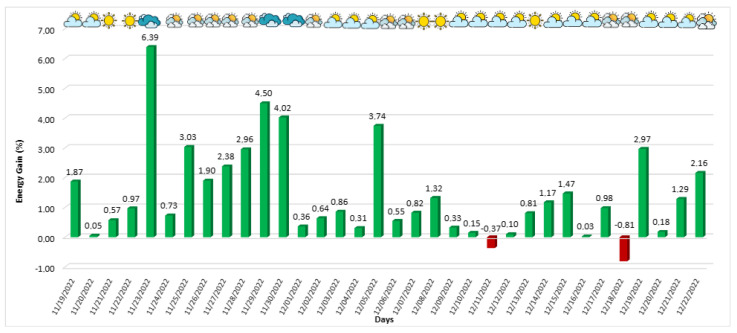
Second version of the developed algorithm’s energy gain in relation to the commercial algorithm.

**Figure 20 sensors-24-03890-f020:**
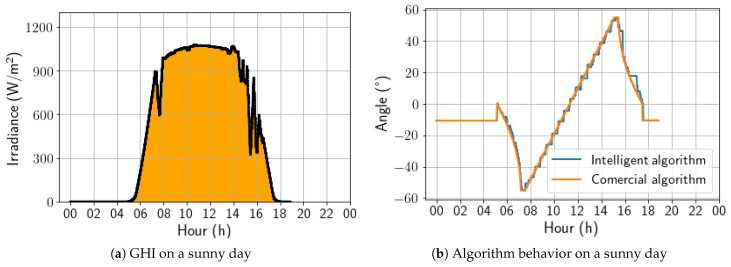
Comparison between intelligent and commercial algorithms for a cloudy day; 24 November 2022.

**Figure 21 sensors-24-03890-f021:**
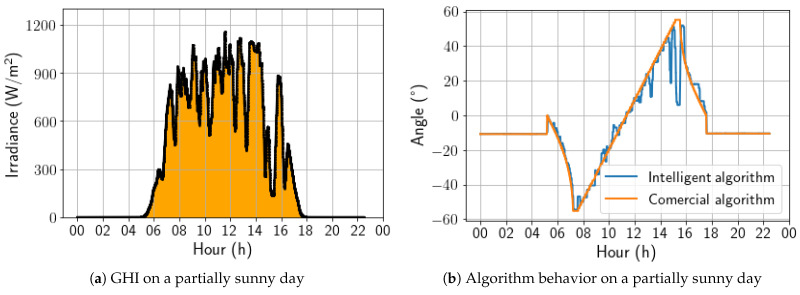
Comparison between intelligent and commercial algorithms on a partial day; 2 December 2022.

**Figure 22 sensors-24-03890-f022:**
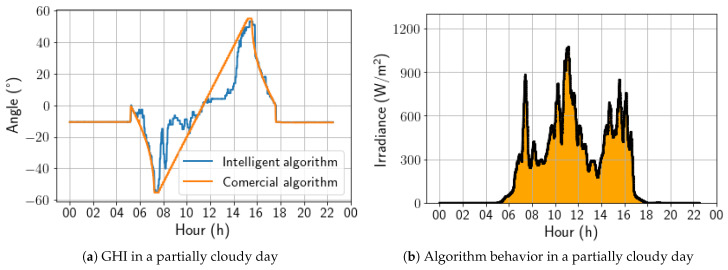
Comparison between intelligent and commercial algorithms for a partially cloudy day; 5 December 2022.

**Figure 23 sensors-24-03890-f023:**
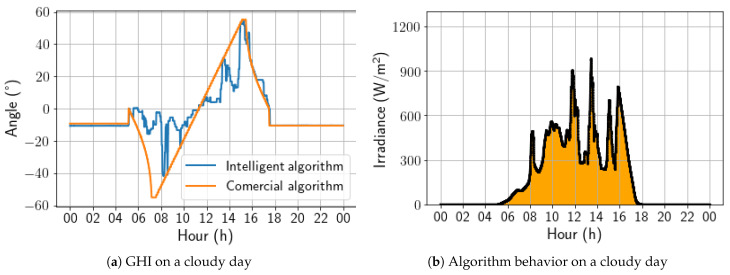
Comparison between intelligent and commercial algorithms for a cloudy day; 5 December 2022.

**Figure 24 sensors-24-03890-f024:**
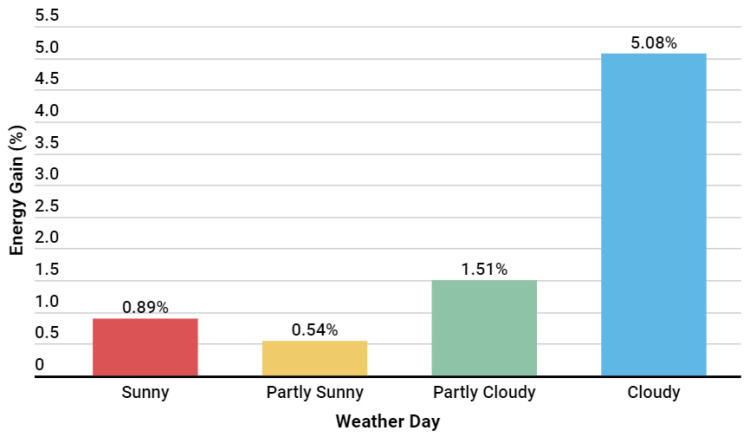
Results by weather situation with the second version algorithm.

**Table 1 sensors-24-03890-t001:** Test environment equipment.

Equipment	Description	Manufacturer
Solar tracker	6 solar trackers rows per inverter	STI Norland
PV panels	60 Astro5Twins 530/535 Wp per tracker row	Astronergy
Inverter	2 Sun2000-185ktl	Huawei
SmartLogger3000	Data logger	Huawei
Weather station	Weather instruments	Campbell

**Table 2 sensors-24-03890-t002:** Test environment equipment communication.

Equipment	Communication Protocol
Solar tracker	Modbus TCP/IP
SmartLogger	Modbus TCP/IP
Weather station	TCP/IP

**Table 3 sensors-24-03890-t003:** Mutual information ranking.

Variable Importance (Mutual Info Regression)	
Variable	Score
Zenith	2.2908
Azimuth	1.9234
Hour_sin	1.1611
GHI	1.3324
Hour_cos	1.1135
Month	0.3843
Temperature	0.2909
Relative Humidity	0.2052
Wind Speed	0.1155
Wind Direction	0.0478
Precipitable Water	0.0363

**Table 4 sensors-24-03890-t004:** Comparison between IA techniques.

Error Metrics			
AI Topologies	MSE	MAE	ME
MLP	5.793	0.862	54.044
LSTM	4.550	0.481	54.123
DT	4.928	0.370	55
RF	4.65	0.333	53

**Table 5 sensors-24-03890-t005:** Variable combination.

Rank	Variable Combination	MSE
1°	Month, GHI, Wind Speed, Zenith, Azimuth, Cos(hour), Sin(Hour)	5.8896
2°	Month, GHI, Relative Humidity, Zenith, Azimuth, Cos(Hour)	5.8898
3°	Month, GHI, Relative Humidity, Zenith, Azimuth, Sin(Hour)	5.8920
4°	Month, GHI, Relative Humidity, Zenith, Azimuth, Cos(hour), Sin(hour)	5.8922
5°	Month, GHI, Wind Speed, Zenith, Azimuth, Cos(hour)	5.8923
6°	Month, GHI, Wind Speed, Relative Humidity, Zenith, Azimuth, Cos(hour)	5.8963
7°	Month, GHI, Wind Speed, Zenith, Azimuth, Sin(Hour)	5.8962

**Table 6 sensors-24-03890-t006:** Weather classification.

Weather Classification	Kt
Cloudy	Kt < 0.35
partially cloudy	0.35 ≤ Kt < 0.55
partially sunny	0.55 ≤ Kt < 0.65
Sunny	Kt ≥ 0.65

**Table 7 sensors-24-03890-t007:** Energy gain.

Algorithm Operation			
Version	N° Days in Operation	Energy Gain (kWh)	Energy Gain (%)
Non-tuned Backtracking	29	273.90	0.73
Tuned Backtracking	34	493.71	1.20

## Data Availability

Restrictions apply to the availability of these data. Data were obtained from the Rio Alto Group, and are available from the authors with the permission of the Rio Alto Group.

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
