# Peer review of "Solar Tracking Control Algorithm Based on Artificial Intelligence Applied to Large-Scale Bifacial Photovoltaic Power Plants"

_sensors, 2024, doi:10.3390/s24123890_

Round 1

Reviewer 1 Report

Comments and Suggestions for Authors

This manuscript presents a novel artificial intelligence (AI)-based algorithm for solar tracker, aimed at optimizing the energy generation of bifacial photovoltaic plants. The research work is deeply discussed in both theory and practice, and verified by actual solar power plant, which shows the improvement and compared with traditional algorithm.

Some Comments and Suggestions:

- The algorithm mentioned in the paper was able to achieve energy gains of up to 7.83% on cloudy days compared to a commercial solar tracker algorithm, which is a remarkable result. However, if the author could provide more detailed data analysis on the algorithm’s performance under different weather conditions, geographical locations and climates, the applicability and scalability of the algorithm can be more comprehensively evaluated.

- The description of the algorithm in this paper is relatively detailed, but the theoretical basis and mathematical model of the algorithm can be further elaborated. If possible, the details of model parameter selection and verification process, such as cross-verification, can be provided to ensure the stability of the algorithm.

- The paper proposed modularization of the algorithm to improve performance during morning and evening backtracking is an innovative point, it is recommended that the authors detail how this optimization affects the overall performance and provide more data support for this.

- In this paper, the results are presented clearly and the discussion is insightful, which suggests adding in-depth analysis of the results, especially why the energy gain is small in sunny days, and how to further improve the algorithm to solve this problem. In the conclusion part, the limitations of the algorithm and the direction of the future improvement are further discussed.

Comments on the Quality of English Language

Overall, English writing is good and please proof read again to correct minor typos.

Author Response

Dear Reviewers,

Our manuscript, "Solar Tracking Control Algorithm based on Artificial Intelligence Applied in a Large-Scale Bifacial Photovoltaics Power Plants," has been revised according to the Reviewers' comments. A deep revision was done to improve the quality of the text. 

We appreciate all the comments and suggestions and consider that they helped us immensely improve the manuscript. We are very grateful for the time taken to review the text.

--------------------

This manuscript presents a novel artificial intelligence (AI)-based algorithm for solar tracker, aimed at optimizing the energy generation of bifacial photovoltaic plants. The research work is deeply discussed in both theory and practice, and verified by actual solar power plant, which shows the improvement and compared with traditional algorithm.

 Some Comments and Suggestions:

  1. The algorithm mentioned in the paper was able to achieve energy gains of up to 7.83% on cloudy days compared to a commercial solar tracker algorithm, which is a remarkable result. However, if the author could provide more detailed data analysis on the algorithm’s performance under different weather conditions, geographical locations and climates, the applicability and scalability of the algorithm can be more comprehensively evaluated.

Answer: The results were from implementing the algorithm in a large commercial plant. The research group is negotiating to validate the solution in different plants in different parts of Brazil. Therefore, at this moment, it will not be possible to present and evaluate the algorithm's performance under different weather conditions. However, this proposal was tested on approximately one month of operation, with different conditions of wind speed, temperature, and irradiance, and presented promising results, reinforcing the solution's innovative characteristic.

2. The description of the algorithm in this paper is relatively detailed, but the theoretical basis and mathematical model of the algorithm can be further elaborated. If possible, the details of model parameter selection and verification process, such as cross-verification, can be provided to ensure the stability of the algorithm.

Answer: To meet the comments concerning this suggestion by the Reviewer, a paragraph has been added to lines 264-274.

3. The paper proposed modularization of the algorithm to improve performance during morning and evening backtracking is an innovative point, it is recommended that the authors detail how this optimization affects the overall performance and provide more data support for this.

Answer: The tracking model proposed in this work performed better than the commercial solution STI-250 from STI Norland. This was primarily due to the use of synergistic information from Month, GHI, Wind speed, Zenith, azimuth, hour Sine, and Hou Cosine, which allowed for the determination of the tracker's optimized angle based on AI.

It should be noted that commercial trackers have predefined angles according to the geographic location and time of day, which limits their optimization when considering climatological data and the seasonality of the time series.

4. In this paper, the results are presented clearly and the discussion is insightful, which suggests adding in-depth analysis of the results, especially why the energy gain is small in sunny days, and how to further improve the algorithm to solve this problem. In the conclusion part, the limitations of the algorithm and the direction of the future improvement are further discussed.

To attend to the requirements established by the reviewer, a paragraph has been added to lines 554-559 and 570-573

Thank you again.

Reviewer 2 Report

Comments and Suggestions for Authors

In this manuscript, the author presents a new artificial intelligence-based algorithm applied to solar trackers that consider bifacial panels by taking into account the unique characteristics of bifacial panels and the physical characteristics of the solar plant. This algorithm is able to increase energy gain during cloudy days and the backtracking period. I would consider accepting this manuscript after the following issues are addressed or clarified.

1. When developing and optimizing algorithms, aside from leveraging diffuse irradiance and improving the backtracking algorithm, how do factors such as direct irradiance, reflected irradiance, and temperature impact energy capture?

2. There are some sections in the manuscript that seem confusing. Section 3.2 contains only one subsection, also similar issues in section 4.2. Could the author provide more clarity or restructure these sections for better coherence?

Author Response

Dear Reviewers,

Our manuscript, "Solar Tracking Control Algorithm based on Artificial Intelligence Applied in a Large-Scale Bifacial Photovoltaics Power Plants," has been revised according to the Reviewers' comments. A deep revision was done to improve the quality of the text. 

We appreciate all the comments and suggestions and consider that they helped us immensely improve the manuscript. We are very grateful for the time taken to review the text.

-------------------

In this manuscript, the author presents a new artificial intelligence-based algorithm applied to solar trackers that consider bifacial panels by taking into account the unique characteristics of bifacial panels and the physical characteristics of the solar plant. This algorithm is able to increase energy gain during cloudy days and the backtracking period. I would consider accepting this manuscript after the following issues are addressed or clarified.

  1. When developing and optimizing algorithms, aside from leveraging diffuse irradiance and improving the backtracking algorithm, how do factors such as direct irradiance, reflected irradiance, and temperature impact energy capture?

Answer:  From the variable selection procedure, a high correlation of the variable’s direct irradiance, reflected irradiance, and temperature was observed with the output variable, tracking angle. This observation was verified mathematically, enabling the development of a more robust model that included these quantities. This procedure was explained in lines 300 to 312.

2. There are some sections in the manuscript that seem confusing. Section 3.2 contains only one subsection, also similar issues in section 4.2. Could the author provide more clarity or restructure these sections for better coherence?

Answer: Section 3 is divided into two large subsections 3.1 MATERIALS (line 165) and 3.2 METHODS: DEVELOPING THE INTELLIGENT ALGORITHM (line 219). Both subsections address different aspects of the methodology, and specifically, section 3.2 discusses the construction of the method proposed in this article. Section 4.2 presents two subdivisions 4.2.1 and 4.2.2